# Why Does Rehabilitation Not (Always) Work in Osteoarthritis? Does Rehabilitation Need Molecular Biology?

**DOI:** 10.3390/ijms24098109

**Published:** 2023-04-30

**Authors:** Adam Zdziechowski, Anna Gluba-Sagr, Jacek Rysz, Marta Woldańska-Okońska

**Affiliations:** 1Department of Internal Diseases, Rehabilitation and Physical Medicine, Medical University, 90-700 Łódź, Poland; 2Department of Nephrology, Hypertension and Family Medicine, Medical University of Lodz, 90-549 Łódź, Poland

**Keywords:** osteoarthrosis, rehabilitation, kinesiotherapy, physical therapy, inflammation regulation

## Abstract

Osteoarthritis (OA) is a common disease among the human population worldwide. OA causes functional impairment, leads to disability and poses serious socioeconomic burden. The rehabilitation offers a function-oriented method to reduce the disability using diverse interventions (kinesiotherapy, physical therapy, occupational therapy, education, and pharmacotherapy). OA as a widespread disease among elderly patients is often treated by rehabilitation specialists and physiotherapists, however the results of rehabilitation are sometimes unsatisfactory. The understanding of molecular mechanisms activated by rehabilitation may enable the development of more effective rehabilitation procedures. Molecular biology methods may prove crucial in rehabilitation as the majority of rehabilitation procedures cannot be estimated in double-blinded placebo-controlled trials commonly used in pharmacotherapy. This article attempts to present and estimate the role of molecular biology in the development of modern rehabilitation. The role of clinicians in adequate molecular biology experimental design is also described.

## 1. Introduction

Osteoarthritis is commonly localized in the knee, hip and hand joints and significantly outnumbers other causes of joint dysfunction and structural degradation. Gradually increasing pain and joint stiffness with range of movement (ROM) reduction are common symptoms of arthrosis. In the majority of cases, the pain urges the patient to search aid from health professionals. Apart from pain, the physiotherapists and medical rehabilitation specialists estimate ROM and physically examine crepitations during movement, muscle strength swelling and excessive liquid presence in the joint. As a routine procedure, the gait pattern is also analyzed. The OA is usually examined via standard X-ray imaging, the inexpensive and most popular method, which in contrast to MRI imaging, is not as effective at displaying the early stages of disease. The clinicians are observing the symptoms and trying to choose optimal treatment based on disturbances revealed by physical examination; joint dysfunction is caused by numerous changes explained by immunologic, genetic and proteomic processes. Molecular biology explains the real background of the processes that lead to clinical symptoms and provides useful tools for revealing the basic mechanisms of kinesiotherapy and physical therapy effects.

It is commonly accepted that arthrosis affects the function and the structure of chondral, bone and synovial tissue together [1]. Even though these tissues possess different mechanical and structural features, there are many common mechanisms leading to joint morphological and functional deterioration [2]. Compared to the bone and synovial membrane, cartilage displays poor regeneration potential [3]. The cartilage in its normal state has no blood vessels and depends on synovia and subchondral connective tissue in metabolic processes. The chondral angiogenesis triggers the cartilage degeneration and increases with OA progression [4]. Chondral matrix is prone to mechanical injuries. Chondral joint surface depletion and smoothness loss lead to acceleration of the mechanical matrix wear. In fact, OA is “all the joint structures disease”, but the cartilage condition and protection prove to be critical for the progression of OA [4].

The macrophages in the joints are present mainly in synovia [5]. The macrophages present M1 or M2 polarization due to the molecular environment in the joint. The macrophages activation (polarization) into M1 form decreases chondrocyte formation from mesenchymal stem cells and deactivates chondral regeneration [5]. The M2 form presents protective properties for joint structures. The dominance of the M1 phenotype increases in OA progression and correlates with clinical state [6,7]. As M1 macrophages play a substantial role in inflammation persistence in OA and rheumatoid arthritis [7], these cells are an attractive aim for novel therapies. According to novel research results, the physical training promotes M2 macrophage polarization [7,8].

Although many studies have been conducted in this field, the arthrosis mechanism is still not well understood. The novel trials draw attention to the molecular background of arthritis [9,10]. The identification of novel molecular paths and factors gives hope for the development of efficient treatment methods which can modify the course of disease. Unfortunately there are many molecular aspects, thus the interaction network seems unclear and complicated. In vitro and in vivo experiments show that the therapy targeted at one specific molecular element of the signaling pathway may provide no clinical effect compared to a placebo or even cause opposite results to the expected [11]. Inflammation, cell senescence and excessive destruction not sufficiently balanced by regenerative processes play a key role in synovial joint degeneration. Arthritis progression is a sequence of parallel but interacting processes. The arthritis course depends on the immune system responsible for stimulation of apoptosis (non-inflammatory programmed cell death [12] and pyroptosis (proinflammatory programmed cell death [12])), oxidative stress, activation of osteoclasts and osteoblasts, enzymatic protein degradation and local pain facilitation. The immune system therefore gives opportunity to inhibit the disease progression or partly reverse arthrosis effects.

Generally, the degenerative diseases occur while the catabolic and apoptotic mechanisms strongly outweigh synthesis processes and cell normal activity including regeneration. OA fulfills this simple pattern. Unfortunately, the molecular biology boom causes confusion because the number of identified and examined proteins, RNA particles and other biologically active substances are rising dramatically. The information management became an autonomic problem which is being solved by bioinformatics [13]. Undoubtedly, immune system dysregulation is responsible for aging processes. The term “inflammaging” (inflammation-induced aging) wisely illustrates this conjunction [14].

## 2. Rehabilitation

It is believed that well-suited training helps to maintain and improve functional state and reduce the OA symptoms [15]. Physical activity should be an integral part of a treatment program in OA therapy [15]. Some positive effects of training on general health condition (maintaining optimal body mass, against sarcopenia [16], and heart failure [17], improving glucose tolerance) raise no doubt, however the influence of training on OA affected joint remains unclear. In fact, neither the type of training nor its appropriate duration and time per week has been established. The results of studies often present contradictory effects also in regard to physical therapy. The crucial unresolved problem of rehabilitation is associated with the lack of high quality studies. The golden standard of modern clinical trials, e.g., blinding is unachievable in studies in the field of rehabilitation and other non-pharmacological treatment methods [18,19,20]. The recent advances in molecular biology show growing potential also in relation to rehabilitation procedures. The recent data shows that OA progresses via different dominant molecular mechanisms as suggested by Ly et al. on a knee OA model [21]. Targeting appropriate mechanisms may prove effective in the therapy. For instance Vassão et al. performed research suggesting an increase in IL-10 in the patient performing an exercise program compared to the control group. The research also showed a lower IL-1β concentration in the exercise group [22]. The training probably plays a minor role among patients with subtypes of OA with a less dominant interleukin dependent molecular pattern. This gives hope of a better understanding and adequate usage of rehabilitation in everyday clinical practice but demands the application of new expensive diagnostic tools. A similar attitude concerning mainly pharmacological methods of OA treatment was demonstrated by Kim et al. [23] in their review.

## 3. Kinesiotherapy-Treatment with Movement

Since the dawn of rehabilitation as a medical discipline after two world wars, kinesiotherapy plays a central role in rehabilitation [24]. As a consequence of accepting the analogy between training and drugs, kinesiotherapy demands appropriate dosage and form [25]. It also determines contraindications and side effects of kinesiotherapy. The training has a great impact on homeostasis and the intra-articular environment and a vast number of articles proving these effects have been published.

The meta-analysis by Alves et al. showed evident and significant changes in the chemokine profile after long distance running [26]. The authors concluded that this exhausting form of sport activity up-regulated proinflammatory IL-6, IL-8, TNF-α serum concentration and caused a parallel increase in the concentration of anti-inflammatory factors IL-1RA (Il-1 receptor antagonist) and IL-10. Long distance running also decreased the IFN-γ and leptin concentration. The changes were more strongly expressed in marathon and ultra-marathon competitors. The research shows the complexity of effort-induced reactions since the final anti-inflammatory or proinflammatory effects are not clear. The training increased the concentration of cartilage oligomeric matrix protein as well as a cartilage matrix damage markers. Balan et al. compared young and old sedentary males with young and old trained cyclists. The results showed positive effects of training on inflammatory markers (Il-8, TNFα) but a lack of significant impact on senescence markers in muscle tissues [27]. This research proves that the tissue senescence does not depend strictly on inflammation (in muscle tissues at least). Nakamura et al. [28] examined a murine model of cartilage destruction. The obtained results suggested that a joint surface load reduction activates cartilage degeneration by NF-κB signaling. The NF-κB consists of several proteins that activate the cell response to stress factors such as cytokines, reactive oxygen species or antigens. An inactive NF-κB is present in cytosol. The activation may occur via classical (dependent on e.g., TNFα, IL-1β, T-cell receptors) or alternative pathways (triggered by adaptive immunity factors). The dysregulation of NF-κB in OA, rheumatoid arthritis and tumor genesis was observed [29]. Jimi et al. suggested that both significantly low and excessively high NF-κB activation leads to OA progression [30]. Deficient NF-κB activity promotes apoptosis, whereas high activity increases inflammation, up-regulation of matrix metalloproteinases and vascular endothelial growth factor (VEGF). The exercise increases the reactive oxygen species (ROS) and reactive nitrogen species (RNS) present in the joints and plasma [31]. The RNS in joints are mainly produced by chondrocytes and macrophages present in large numbers in the synovial membrane [32]. The ROS and RNS, as particles with high oxidative potential, take part in inflammatory reactions and destroy the tissues, e.g., the chondral matrix. On the other hand, the exercise enhances the antioxidative mechanisms. The ROS and RNS activate mitogen-activated kinases (MAPKs), which reorganize the muscle cells genetic activity essential to training-induced adaptive mechanisms [31]. The above-mentioned mechanisms are presented in Figure 1. The data suggests the training should ensure moderate intensity as the exhaustion disturbs the balance and activates destructive catabolic processes including chondrocyte senescence and protease activation [33].

The treadmill exercise shows beneficial effects on inflammation by inhibiting spleen secretion of TNFα, IL-6 and corticosterone increase. The data shows a positive exercise effect on the whole body by general anti-inflammatory action [34]. The review by Kong et al. recapitulates the role of exercise in main molecular processes [31].

### 3.1. Non-Coding RNA

Non-coding RNA (ncRNA) represents a group of RNA not translated into proteins. There are many types of ncRNA including short ncRNA: microRNA (miRNA ranging from 17 to 25 nt), small interfering RNAs (siRNAs) and piwi-interacting RNAs (piRNAs), as well as long non-coding RNA (lncRNA longer than 200-nucleotides). These studies discovered that ncRNA plays a major role in signaling and cellular regulation mechanisms [35].

### 3.2. Micro RNA (miRNA)

The micro RNA was discovered in the 1990s, primarily in procaryotic organisms. Through the decades, these small RNA fragments with different nucleotide sequences were identified and their regulative role became clear. There are more than 50 diseases and developmental disturbances associated with the altered expression of miRNA [36]. Surprisingly, the absence of some miRNA particles causes a better health state (e.g., improved glucose tolerance). miRNA modulates gene expression via guiding Argonaute (AGO) proteins to specific target RNAs which leads in consequence to either RNA degradation or translational repression [37].

Endisha et al. revealed that miR-34a-5p overexpression causes increased cartilage damage, promotes chondrocyte apoptosis and results in clinical deterioration. The miR-34-5p knockdown reduces these negative effects [37]. Ito et al. reported a protective role of miR-455-5p and miR-455-3p on cartilage; these particles hampered hypoxia-inducible factor-2α activity HIF2α (on a mouse model) [38]. HIF2α expression in the cell was induced by low oxygen concentration. HIF2α is activated by creating a complex with sitruin-1 and plays a vital role in activating angiogenesis processes. The miR-455-5p and miR-455-3p synthesis depends on Sox9—the transcription factor responsible for transforming mesenchymal cells into chondrocytes and stimulating chondrocytes normal activity including chondral matrix reparation [39]. The low Sox9 activity in the late stage of OA was observed. Carbonare et al. [40] investigated the impact of half-marathon training on mesenchymal circulating progenitor cells. The authors found that the training increased the Sox9 activity, which resulted in the transformation of progenitors into chondrocytes.

Ding et al. confirmed that miRNA-93 reduced inflammation and protected chondrocytes from apoptosis [41] via interaction with Toll-Like Receptor 4 and preventing NF-κB activation. The protective role of miR146a in osteoarthrosis was characterized by Guan et al. [42]. While working on both human and mice models, the authors discovered that miR146a inhibited the degeneration of cartilage by inactivating Notch 1. The action took place in both post-traumatic and primary OA. Notch 1 is a transmembrane protein influencing cell differentiation and cell communication with other closely located cells. Notch1 presence on chondrocytes located on the cartilage surface was observed and its overexpression characterized chondrocytes in OA. MiR146a has a complementary binding site to Notch1. Notch 1-miR146a interaction leads to down-regulation of Il-1β, IL6, TNF α and attenuates the IL-1β destructive procatabolic activity [42]. Minguzzi et al. reported a reduction in metalloproteinase MMP1, MMP3 and MMP10 expression by Notch1 knockdown, another mechanism of cartilage protection [43]. The results confirmed earlier observations on Notch1 role in OA. At the moment there are no results of research concerning the impact of training on miR-455-3p and miR-455-5p expression.

Horak et al. described the significant changes in extracellular levels of extracellular miR-16, miR-21, miR-93 and miR-222 after performing different forms of training over eight weeks. The volunteers were young male athletic students. The collected data showed changes in circulating RNA (up- or down-regulation). The results are confusing as the four studied miRNAs have a vast influence on many metabolic pathways and receptors [44].

Wardle et al. found differences in plasma miR-16, miR-21, miR-93 and miR-222 levels between the group of endurance-trained and strength-trained athletes [45]. The authors emphasized the sports medicine aspect of the research and did not link the results with OA. In 2021, Stadnik et al. described the regulative role of miR222 (miR-221, miR-21 and miR-27) in cartilage exposed to excessive compression [46]. The increased level of above-mentioned molecules up-regulated tissue inhibitors of metalloproteinase 3 (TIMP3) and cytoplasmic polyadenylation element binding protein-3 (CPEB3). Metalloproteinase 3 cleaves the aggrecan-cartilage matrix basic proteoglycan binding hyaluronic acid and ensures proper cartilage matrix hydration [47]. Thus, the destruction of aggrecan affects the cartilage matrix architecture and function promoting OA development. CPEB3 dysregulation often accompanies occurrence of neoplasm originating from different organs [48]. It is worth mentioning that large compressing forces activate chondral degradation in obese patients.

The circular RNA (circRNA) are commonly present in eucaryotic cells. The circRNA has the C-3 and C-5 endings connected by a covalent bond, which makes it resistant to exonuclease-dependent degradation [48]. Nowadays, the regulative role of circRNA in osteoarthritis has been widely investigated. There have been many circRNA molecules proven to have an important regulative function in aging process regulation (e.g., in Alzheimer disease, heart failure and renal fibrosis) in humans and other organisms. Salzman et al. found that the circRNA sequences represented cell type specificity [49]. Circ_0043947 and circ_0000205 expression increases the presence of Il-1 β and both induce apoptosis and cartilage matrix destruction, however both of these molecules activate different pathways [50,51]. The research suggests that the knockdown of these circRNA may protect the cartilage and reduce the destructive Il-1 β effect.

## 4. Obesity and Adipose Tissue

Exercise and high physical activity prevent obesity. The relation between excessive weight and load the joints must face seems obvious. Similarly, the relation between activity and used-up energy corresponds well with intuition. In fact, these mechanisms do not explain the complexity of obesity-OA interactions.

Collins et al. used a lipodystrophy model of OA in mice. The results suggested the direct influence of adipose tissue on processes engaged in OA occurrence. The implantation of fat pads into lipodystrophic mice surprisingly showed the researchers an observed loss of OA resistance after the adipose tissue implantation. That suggests the direct adipose tissue role in OA progression independent on mass-related effects [52]. The lipodystrophic mice showed OA resistance despite a large fat load in their diet. However, the research does not consider the traumatic effect of the operation itself and emphasizes the adipose effect only. In fact, all joint operations may result in soft tissue adhesions, inflammation and unexpected biomechanical disturbances—any surgical operation has a harmful potential.

### Adipokines

Adipokines are the substances produced by adipocytes with paracrine and endocrine properties [53]. Adipokines are either adipocyte specific proteins (e.g., leptin, adiponectin) or commonly produced by many other cells (e.g., TNFα, IL-1, IL-6). The link between many adipokines and OA was widely described by MacDonald et al. [54]. In 2006, Presle et al. described the difference between serum and synovial distribution of adipokines. The authors proposed the possible role of adipokines in OA [55]. The relation between adipokines and severity of knee OA was confirmed in the study by Calvet et al. [56]. Grffin et al. performed research on overweight mice exposed to a high-fat diet and treated by treadmill exercise. The obtained data suggested a protective exercise effect, although the animals performed exercise with evident knee joint overload. The authors emphasized that the improvement depends on the anti-inflammatory exercise effect hampering proinflammatory cytokines interaction [57].

Leptin, the adipokine discovered in 1994, has a global proinflammatory effect. Obese people usually present a higher level of inflammation markers including increased activity in synovial fluid [54]. As leptin regulates hunger, the hypothalamic regulation via leptin exposure among obese people remains insufficient. Leptin resistance causes excessive food uptake. Leptin decrease occurs after stress. The exercise enhances a response to leptin. In fact, the leptin response regulation figures to be the next mechanism of body mass gain prevention [58]. The OA promoting role of leptin has a multidirectional character as it accelerates cartilage damage via chondrocytes, synoviocytes and osteocyte procatabolic regulation [59]. The meta-analysis by Fedewa et al. showed a positive impact of systematic training on the serum leptin decrease [60]. The obesity leads to higher leptin serum levels which accelerates OA progression. Potentially, the leptin may be used for training efficacy assessment. The resistance training was effective in leptin down-regulation among rheumatoid arthritis female patients [61]. The results obviously cannot be simply extrapolated from OA patients.

Transforming growth factor β2 (TGF-β2) has a multipotent influence on regulative mammalian mechanisms [62]. TGF-β2 acts also as adipokines. Takahashi et al. performed research on mice to investigate the endurance exercise effect on white adipose tissue production. The trial revealed that exercise causes increased TGF-β2 production by adipose tissue. Apart from metabolic benefits (insulin resistance reduction, thermogenesis improvement, better fat rich diet tolerance), TGF-β2 reduces inflammation [63]. TGF-β plays a major role in chondrocyte homeostatic regulation and autophagy, it prevents cartilage degradation.

Martinez-Huenchullan et al. compared constant moderate training with high intensity training of mice treated with a high-fat diet. They found statistical differences in muscle adiponectin release due to training mode and suggested low intensity aerobic exercise is more sufficient at improving the muscle adiponectin level [64]. Orellana et al. suggested a stronger correlation between synovial adiponectin and knee arthrosis functional severity compared with leptin concentration [65]. In fact, the role of adiponectin and other adipokines remains disputable as the results of previous research remain inconsistent [66] (Figure 2).

## 5. Interleukins

Interleukins are the cytokines released by leukocytes responsible for immune reactions [67]. Some interleukins show strong proinflammatory activity while others hamper inflammation and/or improve regeneration and anabolic processes.

IL-1 possesses strong proinflammatory properties [68]. The IL-1 activity increase is observed in both specific and nonspecific immune responses. IL-1 up-regulates production of proinflammatory IL-6 and TNF-α and IL-1 itself. Undoubtedly, IL-1 interacts with chondrocytes determining the NO synthesis increase and proteoglycan production decrease [68]. The chondrocytes have IL-1 receptors which are overexpressed due to inflammatory reaction. The NF-κB pathway activation in chondrocytes demands Il-1 dependent activation and results in chondrocyte senescence [69]. Thus, Il-1 plays a major role in initiating OA. The efforts to successfully treat OA with blocking IL-1 R however failed [70]. The experiments on mice with IL-1 gene coding knockdown showed the disastrous impact of complete Il-1 lack on joints and acceleration of OA [71]. The IL-1 role seems more complex than the proinflammatory factor. The IL-1 acts as an immune system modulator as well.

Il-1 also accelerates genesis of macrophages and lymphocytes, activates reactive oxygen species production and the release of matrix metalloproteinases (catalytic proteins containing zinc atom-MMP). Mainly, the MMP-1 has strong devastating potential in cartilage as this protein destructs type 2 collagen-a basic structural protein of the cartilage matrix [72]. The IL-1 activity demands binding to receptor Il-1R. This fact was used to create artificial antagonist receptor-like Anakinra (approved by FDA for clinical use) to reduce excessive IL-1 activity [73]. Budhiparama et al. in a meta-analysis concluded that IL-1 single nucleotide gene polymorphism determines different knee arthrosis probability [74]. The effect of IL-1 single nucleotide polymorphism was described in 2004, leading to the conclusion that the real effect of resistant exercise depends on Il-1 polymorphism [75]. Further investigation may enable exercise program optimalization using personal gene profile assessment. Genetic evaluation should precede creating a training program and explains why standardization fails to achieve repetitive positive effects.

IL-6 represents strong proinflammatory activity and is one of the senescence-associated secretory phenotype factors together with TNFα. Il-6 dysregulation takes place in autoimmune diseases such as rheumatoid arthritis and juvenile idiopathic arthritis [76]. Tocilizumab, the IL-6 receptor deactivator, is registered for RA, juvenile idiopathic arthritis and COVID-19 treatment. The next substances aimed to inhibit IL-6 activity are being investigated [77]. A high synovial IL-6 concentration in OA is observed. IL-6 presents multiple, both disastrous and protective functions as it activates several signaling pathways [77,78]. Especially signaling via STAT3 seems harmful [78]. The training increases muscular tissue IL-6 production. Il-6 plays an essential role in muscle adaptation to training via promoting osteoblasts into osteoclasts differentiation and osteocalcin production [79]. Il-6 blocking causes training insufficiency in visceral fat mass and decreasing cholesterol serum level [80]. IL-6 blockade reduces free fatty acid (FFA) release during rest and training [81]. FFAs are the major energy substrate for muscles in long-lasting endurance training. Thus, the Il-6 deactivation causes poor training effects. This observation has a major importance as OA leads to secondary sarcopenia and a lack of muscle dependent joint stability. Instability accelerates joint degeneration. The IL-6 inhibition strategy by pharmaceuticals may result in serious biomechanical complications and needs cautious consideration (Figure 3).

## 6. Physical Therapy

Physical therapy uses multiple forms of physical energies, e.g., electric current, magnetic field, mechanical waves, infrared radiation, short and microwaves and tissue cooling (cryotherapy). According to international rehabilitation and physiotherapy societies, guidelines and publications, the physical therapy role in general is being reduced, especially in western countries [82]. There are many profound differences on dosage, duration and other parameters of applied physical stimuli. In different countries, multiple parameters and devices are used which makes performing meta-analyses impossible. Some forms of physical therapy are examined in the context of joint molecular environment.

## 7. Ultrasound Therapy

Ultrasound therapy (US) uses (in rehabilitation physical treatment) devices producing ultrasound wave with power ranging from 0.05 mW/cm^2^ up to 2 W/cm^2^. The frequency applied usually ranges from 0.8 MHz to 3 MHz and the wave energy absorption depth ranges from 1 cm (3 MHz frequency) to 3 cm (0.8 MHz frequency) [83]. In OA the ultrasound wave application takes place in an affected area (the wave penetrates the joint). The ultrasound wave application in some cases forms impulses, in others the wave constantly penetrates the tissue [84]. Among multiple physical therapy methods the ultrasound therapy draws scientific attention, whereas other popular forms of physical treatment (e.g., transcutaneous electric nerve stimulation-TENS and treatment with local and general body cooling-cryotherapy) lack scientifically-proven efficacy and molecular background studies in OA [85,86].

Chinese scientists examined the low intensity pulsed ultrasound (LIPUS) influence on mouse chondrocytes. The results showed the inhibition of proarthrotic vascular endothelium growth factor A (VEGFA) expression which delayed the cartilage loss. LIPUS therapy (average intensity of 30 mW·cm^−2^, frequency of 1.5 MHz, pulse repetition rate of 1 kHz and the on–off ratio of 20% for 20 min) caused the up-regulation of p38 mitogen-activated protein kinase inhibitor [87]. The research on rabbits with surgically triggered knee OA treated with US showed significant changes in synovial fluid protein after US application (continuous mode, 1.5 W/cm^2^, 3 MHz, 10 min per session and five times per week, for two weeks) [88]. Observed changes in proteome had a protective effect, however further investigation should help us understand the whole mechanism of the above-mentioned procedure. Xia et al. examined the LIPUS application (50 mW/cm^2^, on–off ratio of 20%, frequency of 3 MHz) on OA rats chondrocyte co-culture and observed positive effect of ultrasound on mesenchymal stem cell differentiation in cartilage. The authors suggested that LIPUS promotes autophagy (cell self-digestion) with exosome release. Scientists suggest exosome release is essential in transforming mesenchymal cells into chondrocytes [89]. The positive results of long-lasting LIPUS therapy applied on patients with moderate knee arthrosis (4 h daily with 0.13 W/cm^2^ power intensity) on pain reduction and functional improvement were reported by Draper et al. [90]. As low intensity ultrasound does not produce any perceptible for patient sensations, the authors succeeded in performing a placebo-controlled double blind study. This clinical trial does not show the molecular background of the LIPUS effect.

## 8. Magnetic Field Therapy

Therapy with a magnetic field has a long history in physical treatment. Magnetic field influences all the tissues generating electric polarization. The mechanisms underlying magnetic field therapy effects remains unclear, however biophysics and quantum biology show wide a range of possible and proven effects [91]. The magnetic field therapy shows a variety of forms (constant and pulse therapy, magneto-stimulation with μT induction values and 100–1000 Hz basal frequency, magnetotherapy with 1–10 mT induction values and 1–3000 Hz frequency) and various effects as well [92].

Parate et al. used magnetotherapy (10 min 15 Hz impulses amplitude 0.4–5 mT) to investigate its impact on cartilage and mesenchymal stem cells in vitro [93]. The therapy showed a positive effect increase in mesenchymal stem cells transforming into chondrocytes and apoptosis and inflammation inhibition as well. The up-regulation of cell growth factors and anti-inflammatory cytokines (Il-7, Il-11, macrophage colony stimulating factor M-CSF, stem cell factor SCF) was described in the review by Hamid et al. [94]. The cell growth factors (FGF-fibroblast growth factor, TGF-β1 transforming growth factor β1, cell cycle regulator c-Jun) are released intensely as magnetic field increase free K^+^ and Ca^2+^ ions. These changes have a strong effect on cells by enhancing mitotic division and protein synthesis; the anti-inflammatory regenerative processes take advantage. Magnetic field exposure has immunomodulatory properties and the macrophage functional change (anti-inflammatory M2 or proinflammatory M1) depends on applied parameters [95]. The positive effect of pulsed magnetic field was concluded by Yang et al. in a meta-analysis study [92] confirming the positive effect of pulsed electromagnetic field therapy used for OA treatment on pain reduction, stiffness and physical function. The coherent results were presented by Tong et al. [96].

## 9. Laser Therapy

Laser has multiple uses in medicine ranging from surgery, oncology and ophthalmology to rehabilitation. Laser devices are capable of precise dosing electromagnetic wave energy of a certain wavelength. Although laser therapy was invented in the 1960s, its biological activity remains the subject of research until now. In rehabilitation, low level laser therapy (LLLT) with red (wavelength approx. 780 nm) and near infrared (wavelength 808–830 nm) electromagnetic wave application with power output reaching 100 mW [97]. Low level laser therapy does not induce thermal effects, however it activates many biological processes. Wound healing, pain reduction and inflammation suppression are the most common indications for LLLT. Nambi concluded the recent studies performed on a rat osteoarthrosis model in a meta-analysis [98]. The results showed beneficial effects on cytokine profile the reduction in IL-1β, Il-6, TNFα and MMP-13 decrease. Saygun et al. applied low level laser irradiation in vitro on osteoblasts. The irradiation increased the release of basic fibroblast growth factor (bFGF), Insulin-Like Growth Factor I (IGF-I), IGF-I receptor and up-regulated osteoblast activity [99]. However, vast differences in LLLT parameters (wavelength, power density, time of application, energy absorbed) applied prevent reaching some clinical conclusions [100]. Akaltun et al. investigated the high intensity laser therapy (HILT) with IR 1064 nm wavelength laser with a maximal power of 12 W as an addition to the exercise session in knee osteoarthritis versus placebo laser with exercise session. The patients received in total 3000 J of laser energy during 10 sessions. The authors confirmed the effectiveness of HILT addition in pain reduction, range of motion improvement and cartilage preservation [101]. The positive effect of HILT (wavelength 1064 nm energy applied 9.77 J/cm^2^) on IL-10 and vascular endothelial growth factor (VEGF) released from equine mesenchymal stem cells was discovered by Peat et al. [102].

## 10. Further Questions and Few Answers

Clinicians keep on trying to define and choose the finest possible methods of treatment. Molecular biology discovers the genomic and proteomic mechanisms of cell differentiation, aging and function changes. The most interesting results of previous researches are briefly presented in Table 1. Rehabilitation does not easily adopt a double-blinded, randomized trial design and other methods of evaluation must replace the golden standard. The review by Arienti et al. shows the most common shortcomings of publications in rehabilitation [103]. Their article suggests that further progress for increasing rehabilitation research quality is possible, however demands effort. Almost one third of the analyzed rehabilitation studies had random controlled trial design defects.

Molecular biology reveals the web of relations. The final effect of parametric analyzed often does not give the simple answer expected by clinicians (“Does it protect the joint?” “Is high IL-6 level good or bad in synovial fluid?” “Does the training reduce the cartilage destruction?”). However, molecular biology explains why the effects of rehabilitation sometimes seem inefficient. The main agents of OA have many functions and activate diverse mechanisms. The same cytokine may present both protective and destructive effects or trigger different molecular pathways. Ozone (O_3_ triple oxygen atoms particle) demonstrates strong oxidative properties. The oxidative stress in general is a part of the inflammatory reaction. Contrary to intuitive reasoning, the intra-articular application of oxygen and ozone has a protective potential with effect which is comparable to intra-articular hyaluronic acid (HA) application [104,105,106]. The oxygen-ozone intra-articular application presents a low risk of complications yet its efficacy remains controversial. The usage of HA remains a popular method of low invasive treatment, however the cost-effectiveness of HA raises controversy. The inconsistency of clinical trial results in HA clinical efficacy were opposite to discovered positive molecular effects of HA on OA affected joints [107]. The response to many stimuli depends on genetic factors and actual both local and global conditions. Some data suggests osteoarthrosis may have different molecular background. Maybe in the future osteoarthrosis will become a family of diseases with different pathophysiological backgrounds [108].

The computational biology when exploring biological data via artificial neural networks; deep-learning supports efficiently molecular biology [109,110,111]. Some molecular relations cannot be revealed without use of computational techniques [112]. Further AI implementation however must prove its RCT practical viability [109]. Apart from the molecular background of OA, computational biomechanics will support better understanding of OA [112]. Computer-supported techniques are used in exercise evaluation including movement pattern quality [113].

The majority of experiments used murine OA models. Many cellular and signaling mechanisms remain identical among mammals and for many reasons rats and mice replace human models. However, for clinicians the biomechanical differences between humans and murine animals are so enormous that extrapolating murine to human results seems bizarre (“What is the equivalent for mouse treadmill exercise in humans?”). The similarities between murine animals and humans are in fact far more adequate in molecular models compared to biomechanical models. The human and murine lifespan differences influence the molecular processes velocity as well.

On the other hand, other question should be stated. Does molecular biology improve the trial designs discussed with rehabilitation specialists? The clinicians have insight into people’s needs, the real meaning of dysfunction. Rehabilitation, apart from pain intensity aspect, biomechanical evaluation confronts the physical examination with a patient’s needs and beliefs. In many cases rehabilitation cannot reverse the disease effects, however it can compensate the impairment and restore the function. Rehabilitation and physical medicine specialists and physiotherapists usually represent a holistic approach to the disease which makes them valuable partners for geneticists and biologists. The molecule-oriented specialists tend to ignore the fact that the majority of interventions performed during rehabilitation and/or surgical treatment change the biomechanics of OA affected structures. The interdisciplinary teams connecting both biologists and clinicians provide the highest quality trial designs and avoid pitfalls characterizing implicating experimental data into clinical backgrounds. The interdisciplinary research teams or consultants can provide higher quality in further research work.

The main principle of rehabilitation is searching for the interrelation between structure and function. The molecular biology supported by computational biology gives an insight into structure and function in a new useful, inspiring and exciting dimension. Above all, the clinicians can properly estimate the usefulness of molecular biology discoveries to help their patients.

## Figures and Tables

**Figure 1 ijms-24-08109-f001:**
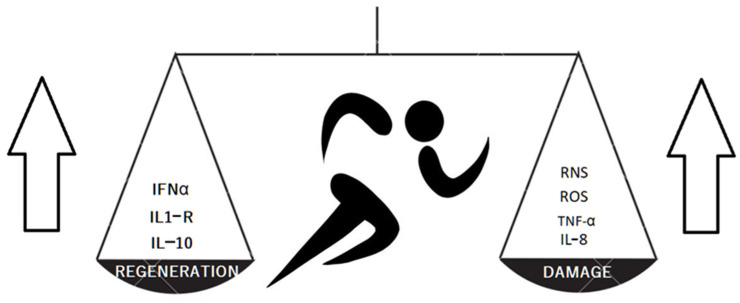
The training results in producing both cartilage protective and damaging factors.

**Figure 2 ijms-24-08109-f002:**
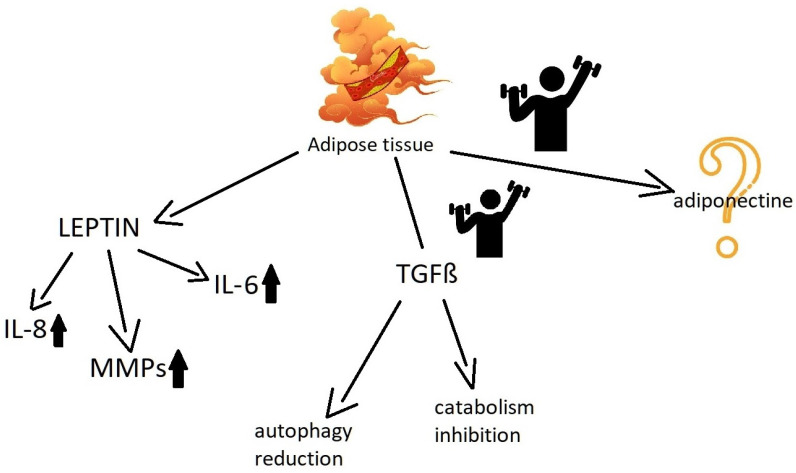
Adipokines possess different properties. Leptin causes proinflammatory changes. Training-induced TGFβ protects the cartilage. The Adiponectine function in OA demands further investigation.

**Figure 3 ijms-24-08109-f003:**
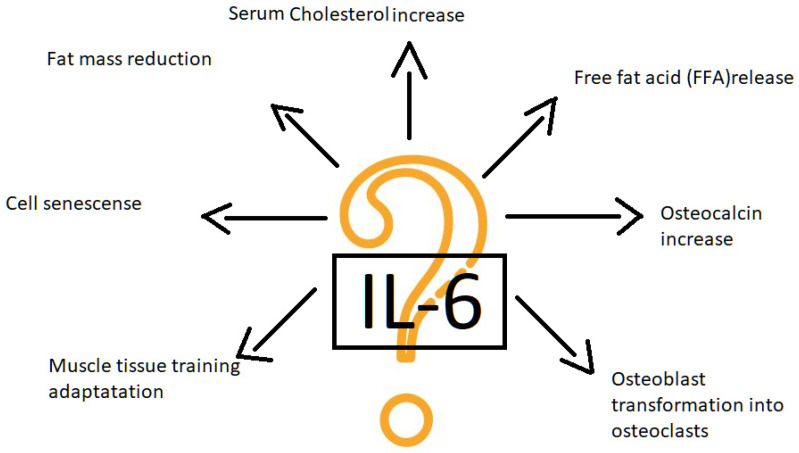
IL-6 shows multiple biological properties. IL-6 the resultant influence on joints demands complex evaluation.

**Table 1 ijms-24-08109-t001:** The significant research results.

Article Type	Human/Animal	Sample Size (Characteristic)	Main Results/Conclusion	Ref.
Research article	Human	6 advanced OA patients	M1 polarized synovial macrophages inhibits mesenchymal stem cells into chondrocytes transformation	[5]
Research article	Human	86 advanced knee OA patients	Synovial macrophage/monocyte M1 polarization correlates with OA symptoms specially stiffness, function and quality of life. Macrophage polarization modulation may be effective in novel treatment strategies	[6]
Review, data synthesis	Human		Suggestion of extracting 4 subtypes of knee osteoarthritis (cartilage degradation driven subtype, bone-remodeling driven subtype, pain driven subtype and inflammation driven subtype). Proposed subtypes present different molecular OA progression background and may demand different treatment strategies	[21]
Randomized blinded study	Human	42 with confirmed knee OA	Exercise program increases a patient’s functional status. Adding IR photobiomodulation to exercise results in increasing serum IL-10 concentration	[22]
Review/meta-analysis	Human	76 research articles	Long distance running increases serum IL-6, IL-1, IL-8, IL-10 and TNFα concentration and reduce IL-2 concentration. Observed effect is stronger in longer distances	[26]
Research article	Rat	15	Reduced dynamic acetabular cartilage load activates Il-6 and MMP3 via the STAT3/periostin/NF-κB axis. Mechanical load is essential for proper cartilage function	[28]
Research article	Human/mouse	37 patients qualified to knee replacement, 34 healthy patients, 120 mice	miR-34a-5p expression is high in plasma and synovium of humans and mice with advanced knee OA. The miR-34a-5p overexpression may be the target for future therapeutic interventions and needs further investigations	[37]
Research article	Human/mouse	21 humans, 23mice	MicroRNA-455-5p and-3p repress hypoxia-inducible factor-2α expression (HIF-2α) and prevent cartilage from HIF-2α procatabolic activity	[38]
Research article	Mouse	36 male mice	MicroRNA-93 protects from apoptosis and inflammation outbreak by TLR4/NF-κB signaling pathway deactivation	[41]
Research	Human	30 young males	Different intense strength training forms performed for 8 weeks change microRNA (miR-16, miR-21, miR-93 and miR-222) plasma concentration	[44]
Research	Human	43	Circular RNA circ_0000205 silencing reduces IL-1β apoptotic activity and cartilage matrix degradation by ADAMTS5 deactivation and miR-766-3p activation	[51]
Research	Human	35 all qualified to knee arthroplasty	Adipokines are produced in joints and their joint activity has no correlation with serum activity. Leptin has higher level in women	[55]
Research	Human	150 Women with advanced KOA with joint effusion	The synovial fluid level of resistin is directly and visfatin level is inversely associated with clinical severity of knee OA	[56]
Review + meta-analysis	Human	72 RCTs	Chronic training causes decrease in leptin and the effect is associated with fat mass loss	[60]
Cross-sectional study	Human	115 women	Synovial adiponectin showed a statistically stronger association with OA clinical progression comparing with synovial leptin	[65]
Review	Human		Il-1 fails as a target in OA therapy	[70]
Research article	Human	24 sedentary men	Inflammatory response strength to resistant exercise depends on Il-1 single nucleotide polymorphism. Potentially polymorphism may be used to training optimalization	[75]
Double-blinded trial	Human	53 abdominal obese	Il-6 blockade (tocilizumab application) results in no effect of endurance training on fat mass loss. IL-6 plays important metabolic role	[80]
Research	Mouse	14	Low intensity ultrasound decrease expression of vascular endothelium growth factor A (VEGFA) via p38 MAPK suppression	[87]
Review + meta-analysis	Rat	188	Low level laser therapy reduces IL-1β, TNF-α, MMP-13. LLLT mechanism has a molecular background	[98]
Research	Human	2402	Adiponectin synovial level is high in advanced knee OA but not in hand joint. The molecular OA pathology differs due to OA localization	[104]

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
