# Peer review of "Why Does Rehabilitation Not (Always) Work in Osteoarthritis? Does Rehabilitation Need Molecular Biology?"

_ijms, 2023, doi:10.3390/ijms24098109_

Round 1

Reviewer 1 Report

The research is interesting and innovative. It needs only some minor changes.

INTRODUCTION: I would recommend inserting a few more bibliographical references to enable the reader to elaborate on the concepts expressed, e.g., lines 38, 40, 50

REHABILITATION CHAPTER:

a) Line 82: please insert bibliographical reference

b) Line 86: This is an important aspect that you outline. Please provide further explanation by comparing with evidence from the literature in some rehabilitation procedures

KINESIOTHERAPY CHAPTER:

a) Line 96: please insert bibliographical reference

b) Lines 112: Can you please include a brief explanation of the mechanism of cartilage degradation related to the NF-kB pathway

PHYSICAL THERAPY CHAPTER:

a)      In addition to the physical therapies you mentioned, it is also important to analyze the role of extracorporeal shock wave therapy and its potential biostimulation properties associated with neovascularization, osteogenesis, and chondrogenesis for both bone and cartilage diseases. For example, I recommend that you analyze and incorporate some of the concepts expressed in these articles that you can cite:

a.       Chen L, Ye L, Liu H, Yang P, Yang B. Extracorporeal Shock Wave Therapy for the Treatment of Osteoarthritis: A Systematic Review and Meta-Analysis. Biomed Res Int. 2020 Mar 18;2020:1907821. doi: 10.1155/2020/1907821.

b.      Sconza C, Anzà M, Di Matteo B, Lipina M, Kon E, Respizzi S, Tibalt E, D'Agostino MC. Extracorporeal shock wave therapy for the treatment of osteonecrosis and bone vascular diseases: a systematic review of randomized controlled trials. Eur Rev Med Pharmacol Sci. 2022 Apr;26(8):2949-2959. doi: 10.26355/eurrev_202204_28626.

Author Response

The authors thank you for your insightful opinion, while improving the article, we followed the instructions provided. All comments contribute to improving consistency and the possibility of deepening the work.

All the bibliographic references indicated in Your comment have been included as noted.

In Poland the ESWT is not treated as therapeutic procedure in rehabilitation by National Health Fund responsible for providing medical service, however the ESWT devices are widely in use, especially in enthesopathy treatment, rarely in OA. As medical rehabilitation specialists we observe major differences in ESWT application, in fact every outpatient clinic uses ESWT devices in its own way. The procedure lacks standardization.

The proper application of ESWT in knee OA treatment suggests Chou et al. It is worth mentioning the study was performed on rats. The mentioned articles by Chen et al. and Sconza et al. present very promising results comparing to alternative treatment methods. There are some interesting research outcomes nicely collected by An et al. and received by Yu et al. revealing molecular mechanisms underlying the positive ESWT results. Our University colleagues however obtained less optimistic results showing superior effects of ultrasound and laser therapy comparing to ESWT. Thus the ESWT subject was omitted in our article.

The articles contributing to molecular background of ESWT:

Chou W.-Y.; Cheng J-H.; Wang C.-J.; Hsu S.-L.; Chen J.-H.; Huang C.-Y. Shockwave Targeting on Subchondral Bone Is More Suitable than Articular Cartilage for Knee Osteoarthritis Int. J. Med. Sci. 2019, Vol. 16 doi: 10.7150/ijms.26659

Yu L.; Liu S.; Zhao z.; Xia L.; Zhang H.; Lou J.; Yang J.; Xing G.; Xing G. Extracorporeal Shock Wave Rebuilt Subchondral Bone In Vivo and Activated Wnt5a/Ca2+ Signaling In Vitro BioMed Research International 2017 https://doi.org/10.1155/2017/1404650

An S.; Li J.; Xie W.; Yin N.; Li Y.; Hu Y. Extracorporeal shockwave treatment in knee osteoarthritis: therapeutic effects and possible mechanism Bioscience Reports (2020) https://doi.org/10.1042/BSR20200926

Reviewer 2 Report

I would like to thank the journal and the editor for inviting me to review this interesting manuscript.

After reviewing this manuscript (IJMS-2232441), my comments are as follows.

# overall opinion 

The manuscript addresses a topic that may be of interest to many readers and is worthy of consideration as a review article; however, I believe it could benefit from further structural and readability improvements.

# Recommendations for improvement

There are countless reviews written on the topic of rehabilitation for osteoarthritis, and while the title of this manuscript is very appealing, the content needs some readability improvements. 

First, I assume that "unmet medical needs in the rehabilitation of OA" is the theme of the entire article. If so, the first thing that should be addressed is "clinical outcomes and pathophysiology that urgently need to be addressed". A compelling definition of these needs to be presented and then the entirety of this manuscript needs to be reorganized around that common outcome and target. 

Second, the manuscript cites several preclinical and clinical studies. If so, a table should be presented to show the data from all the previous studies on the topic at a glance, so the reader can weigh what the authors claim versus the actual data. Along with this, if possible, a scheme should also be presented to show at a glance the molecular mechanism discussed in the manuscript. 

Third, separate paragraphs should be devoted to therapeutic targets that should be targeted in future studies and research methodologies that need to be introduced. This section should be accompanied by a summary problem statement of the flaws and shortcomings of the current study design and performance as a whole. The current manuscript's "10. Further queries.... " section is not sufficient. 

Fourth, in recent years, there have been several molecular biology studies based on computational biology approaches in addition to simple preclinical studies. The manuscript needs to be supplemented with a section addressing the applicability of these new research techniques in OA rehabilitation research and the previous studies in which they have been attempted. 

Fifth, in the end, the two main treatment strategies for OA are structure-based therapies and pain-alleviating therapies. However, DMOADs, which are the backbone of structure-based treatments, have been realized. So, in your opinion, which of these two goals should be the future goal of OA rehabilitation, and can DMOADs even replace the role of structure-based therapies in some cases? I would like to ask for an in-depth discussion to ensure that these themes run through the entire manuscript.

I hope that my views can help improve the manuscript. 

Author Response

The authors thank you for the time devoted to reviewing the work. Each of the comments will be taken into account when improving the work.

First.

The article indicates the role of molecular biology in enhancing rehabilitation effects and explaining the mechanism underlying the clinical outcome of rehabilitation procedures. As a medical rehabilitation specialists we find many aspects of our work unsatisfactory. It is a common conviction among all the physicians, not rehabilitation specialists exclusively. The molecular biology gives hope of better treatment in the future.

The text will be reorganized to define its aim more precisely.

Second. The table will be prepared. We will consider the additional scheme or schemes, however its simplicity and readability presents a challenge.

Third. The section will be improved according to Your comments. We will indicate recommendations for  future projects and common disadvantages of previous works.

Fourth. The computational biology in rehabilitation is a perfect subject for whole monographic article or a book. Undoubtedly the accessibility and high efficiency of modern computers will change our understanding of many processes and problems in medicine and biology. We will shortly introduce this issue.

Fifth

Assuming the DMOAD therapy will offer common access for the OA patients, no side effects and no common contridications the rehabilitation role will change drastically. DMOAD therapy in optimal conditions is theoretically capable for triggering the joint regeneration. In this scenario, the rehabilitation should be aimed to change the biomechanical patterns by remodeling the neural networks  responsible for the movement control.  Some biomechanical changes precede the OA onset and demands correctionas the element of prophylaxis. On the other hand, reasonably lead rehabilitation has very good safety profile and usually does not seriously interact with drugs. It makes rehabilitation attractive method for elder people usually consuming multiple drugs. We believe the rehabilitation and pharmacotherapy may be lead parallely reaching  together synergistic effects.

Reviewer 3 Report

The manuscript of Zdziechowski et al. is a superficial work, lacking consistency from many points of view. First, I have observations for at least two formal aspects: 1. the affiliations are not mentioned, 2. the authors’ role section is empty.

Further, concerning the reviews’ content, I have the following reservations:

1. The title: the term of osteoarthrosis is nowadays is not widely used, as osteoarthritis (OA) replaced it in the scientific terminology. Several clinicians maintain osteoarthrosis, arguing for the lack of classical inflammatory signs. However, inflammation in OA is a low-grade inflammatory process not fulfilling the criteria for the five well-known signs of infective inflammation: redness (rubor), swelling (tumour), heat (calor; only applicable to the body' extremities), pain (dolor) and loss of function (functio laesa). The Osteoarthritis Research Society International defines OA as : “…the most common form of arthritis. It is a disease that affects all the tissues of the joint, including the cartilage, bone, ligaments, and muscles. It can develop in any number of joints, but most commonly affects the knees, hands, and hips.”

2. The manuscript does not have a central concept. For example, the low-grade inflammatory nature of the disease is not emphasized, and some actors of inflammation, chemokines, and pro-inflammatory cytokines, like IL-1 and IL-6 are discussed. The authors argue that anti-IL-1 and anti-IL-6 treatments are not efficient, and question the role of physical training. They do not mention that some NSAIDs, like meloxicam have at least partial efficiency, and do not specify which population group they refer to: those with an initial or those with advanced disease? Obesity-related OA is also associated with a pro-inflammatory phenotype, but there the weight loss, together with cautious physical activity seems to be working.

3. The authors do not focus on several key mechanisms, like cartilage destruction, or synovial macrophage infiltration, which might be improved by rehabilitation. They are not consistent when they enumerate results obtained in experimental and clinical studies. They should keep different section for preclinical and clinical evidence.

4. Several sections, like those dealing with the non-coding RNAs and the physical therapy are very short and superficially treated. The authors mention, that: “In fact the definition of physical therapy remain inconsistent.” Just that is the reason, for which they should have been focused on some, well-defined forms of physical therapy in this review. They introduce the technical aspects of magnetotherapy and laser therapy, but when they cite specific works, they do not mention dose, number of subjects in the experiments, and other circumstances.

5. There are some confusing affirmations in the manuscript, like:

a. „The training results in producing both cartilage protective and damaging factors. Thus the role of physical activity remains unequivocal.”(?)

b. “As IL-6 shows multiple biological properties, its activity changes may reveal unpredictable effects for joints.” IL-6 has at least two signaling mechanisms, and, depending on the environment, may exert both pro-inflammatory and anti-inflammatory roles.

6. There are improper questions without an answer in the last section, like: “Does molecular biology needs rehabilitation specialists?” Maybe, the authors wanted to write reversely, if rehabilitation therapy needs molecular biologists? According to the OARSI, “OA is considered a chronic (long-lasting) disease and other than joint replacement surgery there is presently no cure. There are, however, treatments that can reduce pain, improve function, and in some instances delay the progression of the disease.”

This means that both molecular biologist (in research concerning diagnostic, staging and monitoring biomarkers, and in genetic investigation) and rehabilitation specialist have their well-defined role in the management of this chronic disease.

References

1. https://oarsi.org/what-osteoarthritis

2. Tanchev P.: Osteoarthritis or Osteoarthrosis: Commentary on Misuse of Terms. Reconstructive Review 2017, 7(1):46. http://dx.doi.org/10.15438/rr.7.1.178

Author Response

The authors would like to thank for the insightful and valuable comments that allow for the improvement of the work in terms of content and structure.

The language text correction has been performed.

The afiliations and  authors role will be mentioned.

  1. The authors agree with the Reviewer on the etiology of the disease and osteatrhrosis has been changed to osteoarthritis in the title and content of the paper.
  2. The authors' intention is not questioning the physical activity as a form of treatment. The role of training changes as OA progresses - it will be emphasized in the article. The inflammatory background of OA is clear and the excess adipose tissue activity is presented in the text. The NSAID's are commonly used worldwide and the rehabilitation does not reduce their activity. The adequate rehabilitation does not possess some of the NSAIDs limitations - contrindication for patients using NOACs, gastric and duodenal inflammation risk or need of caution in coronary disease and other cardiovascular diseases.
  3. The mentioned mechanisms will be added to the manuscript.
  4. The cited data will be presented more precisely.
  5. The affirmations will be edited.
  6. The authors does not question the role of molecular or any other specialists in OA management. We intend to show that interdisciplinary approach gives better results in designing the trials and interpreting the outcomes. Rehabilitation demands interdisciplinary and holistic approach and we do believe that collaboration of clinicians and basic sciences specialists will result in faster and more efficient developement of diagnostics and treatment. The section will be reedited.

Round 2

Reviewer 2 Report

I believe that the authors have sufficiently reflected my views, and I am therefore of the opinion that this manuscript can be published as an informative article on the treatment of OA. 

Author Response

~